# Isostructural doping for organic persistent mechanoluminescence

Zongliang Xie[1,2], Yufeng Xue[2], Xianhe Zhang [2], Junru Chen[2], Zesen Lin[1,2] & Bin Liu [1,2] ✉

Mechanoluminescence, featuring light emission triggered by mechanical stimuli, holds immense promise for diverse applications. However, most organic Mechanoluminescence materials suffer from short-lived luminescence, limiting their practical applications. Herein, we report isostructural doping as a valuable strategy to address this challenge. By strategically modifying the host matrices with specific functional groups and simultaneously engineering guest molecules with structurally analogous features for isostructural doping, we have successfully achieved diverse multicolor and high-efficiency persistent mechanoluminescence materials with ultralong lifetimes. The underlying persistent mechanoluminescence mechanism and the universality of the isostructural doping strategy are also clearly elucidated and verified. Moreover, stress sensing devices are fabricated to show their promising prospects in high-resolution optical storage, pressure-sensitive displays, and stress monitoring. This work may facilitate the development of highly efficient organic persistent mechanoluminescence materials, expanding the horizons of next-generation smart luminescent technologies.

Mechanoluminescence (ML) materials, capable of spontaneous light emission triggered by external mechanical stimuli (rubbing, scratching, pressing, shaking, etc.), have attracted widespread attention owing to the merits of environment-friendly excitation mode and real-time detection capabilities[1–3]. As a distinctive kind of smart luminescence, ML establishes a direct link between mechanical stimulation and luminescent responses, promising an array of potential applications in stress-sensing, pressure-sensitive lighting and display, wearable devices, artificial intelligent skin, information encryption, and anti-counterfeiting[1–7]. ML phenomenon was initially documented in the early 1600s by Francis Bacon[8]. To date, by virtue of stable and repeatable ML performance, a range of inorganic ML materials have been broadly harnessed for producing sensing devices, electronic signatures, and many other commercial applications[2,3,9]. Compared with inorganic counterparts, organic ML materials offer additional advantages, such as ease of modification, low toxicity, flexibility, solubility, and abundant luminescence characteristics[7,9,10]. In recent years, a notable succession of vibrant organic ML materials,

encompassing derivatives of carbazole[7,11,12], arylethylene[4,6,13], phenothiazine[14–16], triphenylamine[17,18], and acid imide[19,20], have been effectively developed. Several design principles have been proposed for organic ML materials as well[4,13,14,21,22]. However, most existing organic ML materials featured instantaneous luminescence with short emission lifetimes in the nanosecond range[4,7,11–19]. The ML signals can only be detected by specialized spectrometers, impeding their practical applications.

Organic afterglow materials, characterized by ultralong lifetimes exceeding tens of milliseconds or even seconds, have been developed using various strategies such as crystallization-inducement[23,24], H-aggregation[25,26], polymerization[27–29], cocrystallization[30,31], host–guest doping[32–35], and numerous other methodologies[36,37]. If afterglow materials can be excited by mechanical stimulation, it would present an effective approach to overcome the limitations arising from instantaneous ML. The detection windows could be prolonged with an extended lifetime, while minimizing background interference during the sensing process. However, it remains a formidable challenge to achieve

[1]Institute for Functional Intelligent Materials, National University of Singapore, Singapore, Singapore. [2]Department of Chemical and Biomolecular Engineering, National University of Singapore, Singapore, Singapore. ✉e-mail: cheliub@nus.edu.sg

persistent ML (*p*ML) in purely organic molecules. Up until now, almost all the reported organic afterglow materials were merely excited by optical means, but not by mechanical forces[23–41]. Simultaneously, considering the piezoelectric effect and increasing the proportion of long-lived phosphorescence components during molecular design remains challenging. Thus, a facile and universal strategy for the precise design of organic *p*ML materials to meet high-performance prolonged sensing capabilities is in urgent demand.

Leveraging the rigid host to suppress non-radiative transition and stabilize triplet excitons, significant progress has been made in host–guest doping for achieving organic afterglow materials recently[32–35,38–43]. Particularly, for those doping systems with isomeric host and guest molecules, suitable energy-level disparities within the host–guest pairs could be achieved to facilitate effective charge transfer and separation during photoirradiation[35,39–41]. Such an excited-state process has some similarities to that in organic ML, which generally involves spatial charge accumulation and separation on the fractured surface induced by piezoelectricity, and then recombination to emit light[1,10]. Based on this premise, if further structural modifications are applied to the piezoelectric host and guest molecules to enhance the ratio of the afterglow components, it could lead to *p*ML.

Herein, we present the isostructural doping strategy for achieving organic *p*ML, where hosts and guests demonstrate similar structures. Through isostructural doping with structural modification (as detailed in "Methods" and shown in Fig. 1a), we successfully obtained a series of *p*ML materials exhibiting diverse phosphorescence lifetimes (ranging from 18.8 to 384.1 ms) and multiple *p*ML colors (green, yellow, and orange) at room temperature. Detailed photophysical investigations, single-crystal analyses, and theoretical simulations were carried out to unveil the intrinsic mechanism of *p*ML. Notably, the isostructural doping strategy demonstrated excellent universality as we adopted diverse host and guest molecules within the design principle, all of which yielded anticipated *p*ML outcomes. Furthermore, we fabricated simple stress-sensing devices through the melt-casting method and conducted preliminary investigations into the application of *p*ML in this context.

## Results

9-Phenyl-9H-carbazole (PC) and its analogs exhibit promise in the construction of organic ML due to their inherent piezoelectric skeleton and special crystal structures[12,44,45]. Therefore, we devised a series of piezoelectric molecules, namely PC, BPC, and BCPC, along with their isostructural counterparts, PB, BPB, and BCPB (Fig. 1b), to establish a range of host–guest doping systems with molar percentage of 1%. All these molecules were synthesized via a simple one-step C-N coupling reaction. As depicted in Fig. 2a, these three host materials only exhibited short-lived blue fluorescence upon excitation by UV light or mechanical stimuli. Nevertheless, all isostructural doping systems displayed sustained orange luminescence that persisted several seconds after the cessation of 365 nm UV light, characterized by ultralong lifetimes exceeding 100 ms (Fig. 2b, c). The delayed PL spectra of these three doping systems matched well with that of their respective guest species (PB, BPB, and BCPB) in vibrational structure and emission wavelength (~ 565, 613, and 673 nm), aligning closely with the phosphorescence attributed to the benzoindole moiety, as documented in Fig. 2d and Supplementary Fig. 2. Structural modifications to both the host and guest molecules, involving the introduction of the bromine atom and cyano group, can notably boost the afterglow emission of the doping materials. In comparison with PC&PB, the proportion of phosphorescence components of the other two doping materials, BPC&BPB and especially BCPC&BCPB, experienced substantial enhancements, leading to an increase of over 90-fold in the ratio of phosphorescence component ($R_{Phos}$), as detailed in Supplementary Table 2. The phosphorescence quantum yields ($\Phi_{Phos}$) of BCPC&BCPB (1.92%) also exhibited a significant enhancement compared with PC&PB (<0.08%), as shown in Fig. 2b. Therefore, BCPC&BCPB demonstrated exceptional room temperature *p*ML with sustained orange luminescence (with peaks at 565, 613, and 674 nm) persisting after the cessation of mechanical stimulation (Fig. 2d and Supplementary Fig. 4), whereas PC&PB and BPC&BPB only exhibited instantaneous ML under the same conditions, despite their capability for afterglow emission upon photoexcitation.

To clearly elucidate the isostructural doping strategy and *p*ML mechanism, it is imperative to delve into the underlying processes governing the high-efficiency afterglow and ML formation. The cyano

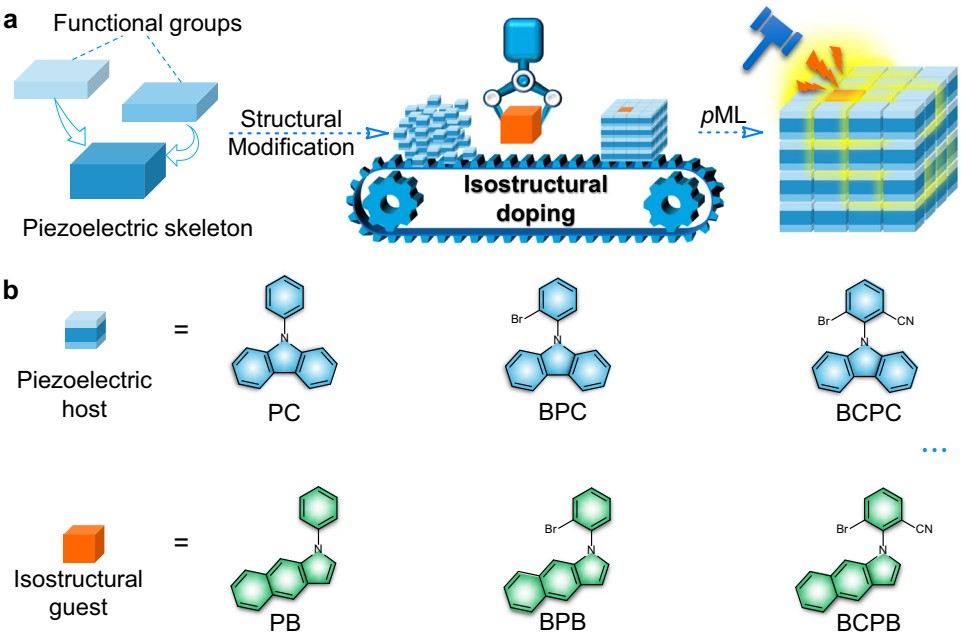

**Fig. 1 | Rational design of organic *p*ML materials. a** Schematic illustration of isostructural doping strategy. **b** Chemical structures of the piezoelectric hosts and the relevant isostructural guests according to the design strategy.

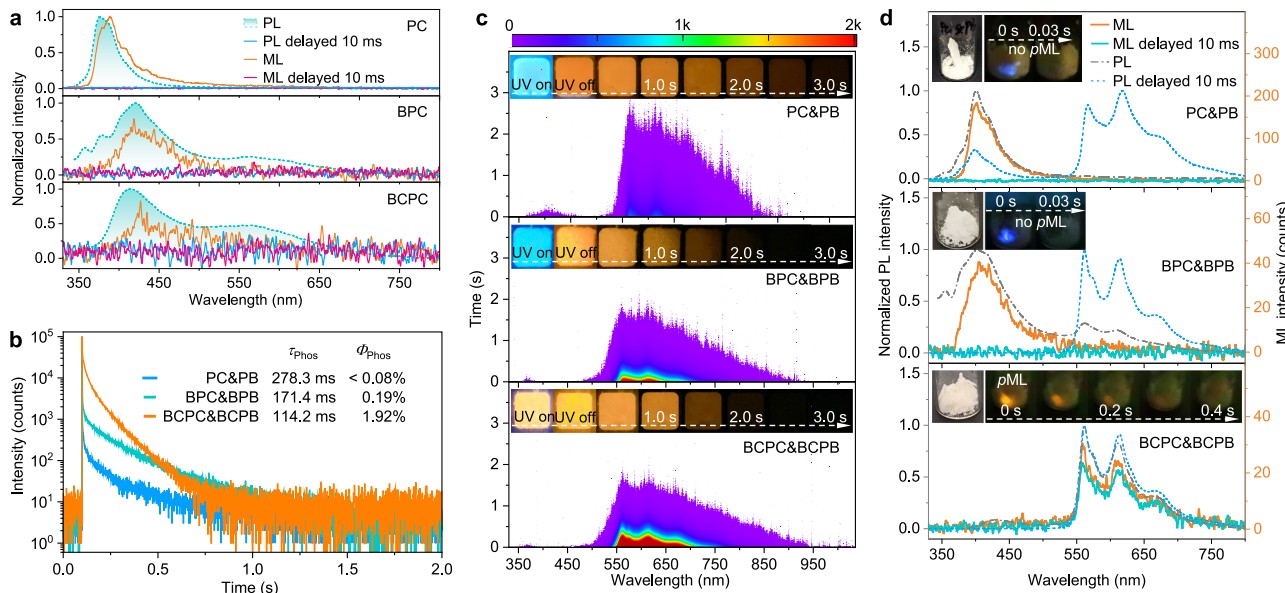

**Fig. 2 | Photophysical properties of the host, guest molecules, and iso-structural doping systems at room temperature. a** Normalized photo-luminescence (PL, steady state and delayed 10 ms) and mechanoluminescence (ML, prompt and delayed 10 ms) spectra of the host (PC, BPC, and BCPC) crystalline powders. **b** PL decay profiles of the afterglow emission at 565 nm for PC&PB, BPC&BPB, and BCPC&BCPB ($\lambda_{ex} = 365$ nm). **c** Time-resolved emission spectra (TRES) mapping of PC&PB, BPC&BPB, and BCPC&BCPB. Inset: Luminescent photographs of the isostructural doping systems under UV on and UV off. **d** PL (steady state and delayed 10 ms) and ML (prompt and delayed 10 ms) spectra of PC&PB, BPC&BPB, and BCPC&BCPB. Inset: ML images of the isostructural doping systems.

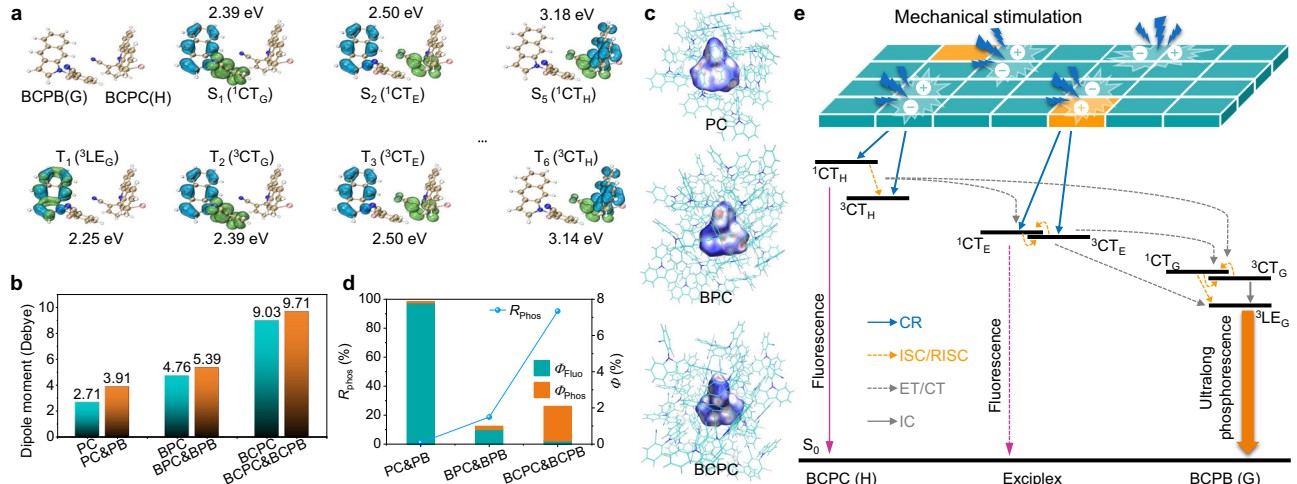

**Fig. 3 | Mechanism investigations of $p$ML in isostructural doping systems. a** TD-DFT calculation results of excited states of BCPC&BCPB optimized from BCPC single crystal: the isosurface maps of electron-hole density difference (blue and green isosurfaces correspond to hole and electron distribution, respectively). **b** Diplole moments of the dimers of host molecules (PC, BPC, and BCPC) and their corresponding doping systems (PC&PB, BPC&BPB, and BCPC&BCPB). **c** Hirshfeld surface analysis of PC, BPC, and BCPC. **d** The ratio of phosphorescence component ($R_{Phos}$), fluorescence quantum yields ($\Phi_{Fluo}$) and phosphorescence quantum yields ($\Phi_{Phos}$) for these three isostructural doping systems. **e** Jablonski diagram showing the proposed mechanism of organic $p$ML in isostructural doping systems with BCPC&BCPB as an example. CR charge recombination, ISC intersystem crossing, RISC reversed-intersystem crossing, ET energy transfer, or CT charge transfer (CT) between different singlet (or triplet) excited states, and internal conversion (IC).

group, serving as an electron-absorbing moiety, can effectively polarize the host and guest molecules and enhance their charge transfer (CT) characteristics. Time-resolved emission spectra (TRES) of BCPC, BCPB, and BCPC&BCPB at the nanosecond timescale (see Supplementary Fig. 5) were recorded to investigate the CT characteristics of the isostructural doping system. The introduction of the cyano group led to evident CT properties in BCPC and BCPB, manifesting as a redshift in their transient emissions from 456 and 497 nm to 483 and 542 nm, respectively. Notably, both the transient emission wavelength and redshift of BCPC&BCPB (shifting from 475 to 493 nm) are different

from those of BCPC and BCPB, individually. This difference, coupled with the unstructured and broadened emission bands, indicated the formation of a CT exciplex in BCPC&BCPB. Furthermore, time-dependent density functional theory (TD-DFT) calculations of excited states of BCPC&BCPB were performed to support the above experiment results. The singlet and triplet CT excited states of BCPC&BCPB exciplex, as presented in Fig. 3a and Supplementary Fig. 6, were situated at elevated energy levels, precisely at S$_2$ ($^1$CT$_E$) and T$_3$ ($^3$CT$_E$). Notably, they fell between the intramolecular CT excited states of the host (S$_5$ ($^1$CT$_H$) and T$_6$ ($^3$CT$_H$)) and guest (S$_1$ ($^1$CT$_G$)

and $T_2$ ($^3CT_G$)) molecules, in consistent with the findings derived from TRES results. Thus, the host–guest CT exciplex can provide a suitable energy-level platform to facilitate more efficient transitions of excited-state excitons from host to guest, thereby promoting high-efficiency afterglow formation. PC&PB and BPC&BPB also displayed similar patterns to that observed in BCPC&BCPB (Supplementary Figs. 7 and 8). In addition, as depicted in Supplementary Figs. 6–8 and detailed in Supplementary Tables 3–5, the pronounced CT characteristics of BCPC&BCPB contributed to narrowing the energy gaps between the lowest triplet excited state ($T_1$) and higher excited states ($S_n$ or $T_n$, where $n \geq 2$). This reduction in energy gaps could further facilitate the transition of excited excitons, ultimately resulting in an enhanced phosphorescence efficiency when compared to that observed in PC&PB and BPC&BPB.

Single-crystal analysis of these three host molecules and theoretical simulations based on their crystal structures were carried out to clarify the $p$ML mechanism. As illustrated in Supplementary Tables 6–8, single-crystal structures of these host molecules belong to orthorhombic and monoclinic systems, characterized by non-centrosymmetric polar space groups: $Fdd$2 for PC, $Iba$2 for BPC, and $P2_1$ for BCPC. Generally, the piezoelectric property, which is influenced by molecular dipole moments and the non-centrosymmetric spatial arrangement of crystals, constitutes a critical factor in organic ML formation. By employing structural modification and isostructural doping strategies to polarize the host–guest pairs, these three doping systems demonstrated a significant enhancement in dipole moments, progressing from PC&PB, BPC&BPB to BCPC&BCPB (Fig. 3b and Supplementary Fig. 13). Subsequently, the special packing modes of these crystalline host matrices significantly augmented their total dipole moments and heightened the probability of ML formation. However, the ML disparities between PC&PB, BPC&BPB, and BCPC&BCPB revealed that achieving $p$ML is not merely a straightforward sum of afterglow and ML. It also necessitates an elevation in the $R_{Phos}$ within the overall luminescence. In these cases, certain key factors, including the intersystem crossing efficiency between $T_1$ and $S_0$, as well as the suppression of non-radiative transitions, all need careful consideration. In Supplementary Figs. 14–17, it is evident that the crystal voids within single crystals of PC, BPC, and BCPC exhibited a very slight decrease (from 34.4 to 33.2 and 32.1%), meanwhile, a noteworthy enhancement in intermolecular interactions was observed. PC dimer exhibited relatively relaxed C–H···π interactions (with distances of 2.71–2.83 Å). Conversely, dimers of BPC and BCPC displayed tighter C–H···π interactions (with distances of 2.54–2.71 Å and 2.67–2.75 Å, respectively). Furthermore, BCPC dimer revealed additional intermolecular interactions, including C–H···N (2.63 Å) and C≡N···Br (3.31 Å). These findings suggest that the incorporation of the bromine atom and cyano group facilitates the effective embedding of isostructural guest molecules into host matrices, concurrently augmenting intermolecular interactions. It is reinforced by the results of Hirschfeld surface and fingerprint plot analyses performed on these host single crystals, which consistently indicated a substantial enhancement of intermolecular interactions resulting from structural modification. This enhancement was characterized by a progressively intensified region of intermolecular interactions (highlighted in red in Fig. 3c) and an increased proportion of interactions involving bromine atom, cyano group, and other moieties (Supplementary Figs. 18–20). In a similar vein, we can infer that when these isostructural guest molecules are incorporated into the host matrices, the host–guest pairs will likewise undergo strengthened intermolecular interactions, which can mitigate the non-radiative losses of triplet excitons and enhance the phosphorescence efficiency. Furthermore, the introduction of bromine atom and cyano group can also enhance the spin-orbital coupling (SOC) between $T_1$ and $S_0$ (Supplementary Fig. 21) due to the heavy atom effect and heteroatom effect (affording extra n orbitals for hybrid n-π* transition) respectively, thereby promoting

phosphorescence in the doping materials. Due to these synergistic effects, the $R_{Phos}$ of BCPC&BCPB (up to 91.8%) far exceeded that in the other two doping systems (<1.0% for PC&PB and 18.7% for BPC&BPB, respectively, as shown in Fig. 3d), leading to the dominance of phosphorescence in the luminescent process and, consequently, the attainment of $p$ML. It is worth mentioning that BPC&BPB also demonstrated $p$ML at 77 K, as illustrated in Supplementary Fig. 22. Moreover, BCPC&BCPB demonstrated an additional emission band (at 500–550 nm) in its $p$ML spectra at 77 K, which was attributed to the phosphorescence of BCPC induced by suppression of non-radiative transitions. This further underscores the importance of effective inhibition of non-radiative transitions to enhance the phosphorescence efficiency and achieve $p$ML.

It is noteworthy that the instantaneous ML emitted by the host species could be detected at the moment of mechanical stimulation applied to the crystalline powders of these isostructural doping materials (as depicted in Supplementary Figs. 23 and 26). Furthermore, the ultralong phosphorescence component continued to intensify even after the cessation of mechanical stimulation, while the blue instantaneous ML of the host species vanished. These findings suggested that the host species could also contribute to the generation of $p$ML. To provide a clearer understanding of the isostructural doping strategy with structural modification and to delve deeper into the mechanism underlying $p$ML generation, we have summarized the Jablonski diagram depicting the proposed photophysical processes corresponding to BCPC&BCPB, as presented in Fig. 3e. Upon mechanical stimulation, charges accumulated at the fractured surfaces of the doping system and subsequently recombined, generating excitons at the CT excited states of the BCPC&BCPB exciplexes and host species. Moreover, the formation of the BCPC&BCPB exciplex provided a suitable energy-level platform ($^1CT_E$ and $^3CT_E$) between the host ($^1CT_H$ and $^3CT_H$) and guest species ($^1CT_G$ and $^3CT_G$), facilitating more efficient transitions of the excited excitons from higher host and exciplex excited states to the lowest triplet state ($^3LE_G$), ultimately emitting high-efficiency ultralong phosphorescence to achieve $p$ML.

To demonstrate the universality of the isostructural doping strategy with structural modification and exploit organic $p$ML materials, we designed two additional guest molecules ($b$-BCPC and $c$-BCPC, Fig. 4a) with analogous structures and the same elemental composition as BCPC. Both BCPC&$b$-BCPC and BCPC&$c$-BCPC exhibited significantly high $R_{Phos}$ within the overall luminescence, exceeding 91%. As shown in Fig. 4b, c and Supplementary Figs. 24 and 25, BCPC&$b$-BCPC exhibited promoted afterglow emission with a lifetime of 233.7 ms and a greatly enhanced $\Phi_{Phos}$ of 11.9%. Hence, an impressive $p$ML instance featuring a persistent yellow emission (with peaks at 543, 591, and 646 nm, Fig. 4d) lasting for ~1 s after the cessation of mechanical stimulation was readily observed at room temperature. Compared with BCPC&$b$-BCPC, BCPC&$c$-BCPC exhibited green $p$ML characterized by sustained emission peaking at 499, 537, and 581 nm upon mechanical stimulation (Fig. 4e), with a shorter phosphorescence lifetime (18.8 ms) and a lower $\Phi_{Phos}$ (6.1%).

Besides modifying guest molecules, piezoelectric molecules with analogous structures can also be employed as the host matrixes within this design principle. In this context, a phenothiazine oxide derivative, BCPSO, sharing similarities with BCPC in appearance but possessing different elemental composition, was purposefully devised as a new host molecule. Similarly, BCPSO single crystal belonged to the $P2_1$ non-centrosymmetric polar space group, which was conducive to ML formation. By isostructural doping, both BCPSO&$b$-BCPC and BCPSO&$c$-BCPC reached very high $R_{Phos}$ of nearly 95% with ultralong lifetimes of 384.1 ms and 319.4 ms, and afforded decent $\Phi_{Phos}$ of 7.3% and 3.5%, respectively. Yellow and green $p$ML can also be achieved in these two doping systems at room temperature (Fig. 4f, g). Moreover, in line with this design principle, we developed an isostructural guest molecule tailored for 2BCzA, a piezoelectric compound previously reported in

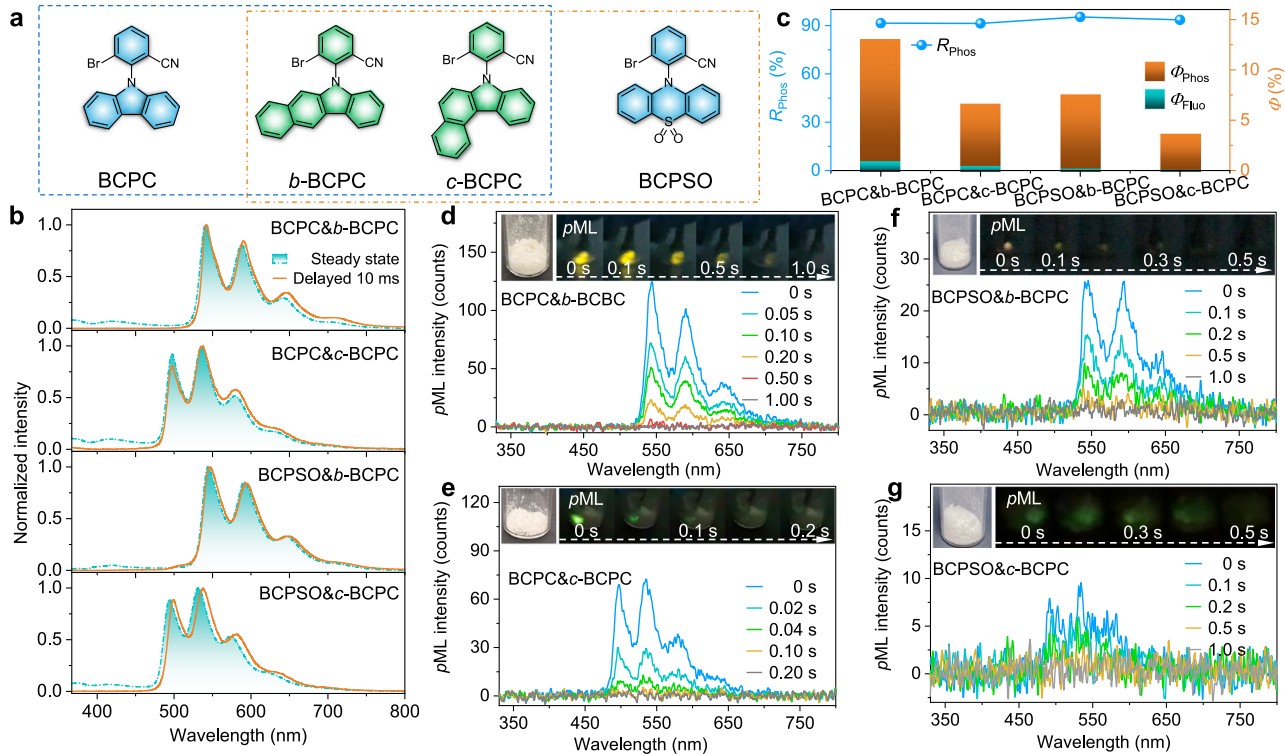

**Fig. 4 | The general strategy of achieving *p*ML with diverse isostructural host and guest molecules. a** Chemical structures of the isostructural host and guest molecules. **b** Normalized PL (steady state and delayed 10 ms) spectra of BCPC&*b*-BCPC, BCPC&*c*-BCPC, BCPSO&*b*-BCPC, and BCPSO&*c*-BCPC. **c** $R_{Phos}$, $\Phi_{Fluo}$, and $\Phi_{Phos}$ for these four isostructural doping systems. **d**–**g** TRES of *p*ML for BCPC&*b*-BCPC (**d**), BCPC&*c*-BCPC (**e**), BCPSO&*b*-BCPC (**f**), and BCPSO&*c*-BCPC (**g**). Inset: *p*ML images of these four doping systems.

the literature[22], and successfully elicited the anticipated *p*ML property in the corresponding doping system 2BCzA&*b*-CzA, as shown in Supplementary Fig. 28. Accordingly, the experimental results above validate the universality of our proposed design strategy. In addition, it has also been demonstrated that multicolor *p*ML with green, yellow, and orange afterglow emission can be achieved by modifying the isostructural guest molecules to tune the energy level of radiative triplet state $^3LE_G$.

Taking advantage of the eco-friendly excitation mode and afterglow emission with prolonged sensing capabilities, we explored potential applications in *p*ML lighting, displays, and sensing as a proof of concept. The prototype display and sensing devices (Fig. 5a) with a series of doping materials based on BCPC were developed using the melt-casting method. In Fig. 5b, an array of high-resolution multicolor afterglow patterns, featuring images such as fish, bird, amusement park, and more, were readily attainable using the mask technology, highlighting the considerable potential for high-resolution optical storage. Thanks to their highly efficient afterglow triggered by mechanical stimulation, these *p*ML-emitting crystals can illuminate dark environments, presenting an eco-friendly light source for pressure-sensitive lighting (Supplementary Fig. 29). Also, a pressure-sensitive display was accomplished by employing a mask pattern on the *p*ML panel and pressing. As depicted in Fig. 5c, it resulted in a vibrant *p*ML displaying the information I♥NUS. Furthermore, stress visualization and monitoring can be achieved and demonstrated in Fig. 5d. When the sensing device was subjected to mechanical stress, the stressed region would emit afterglow for straightforward detection. Meanwhile, during the imposition of stress, *p*ML formed and persisted along the stress trajectory, opening possibilities for applications in stress distribution monitoring and material damage detection.

In conclusion, we have successfully presented a valuable strategy for the achievement of organic *p*ML materials characterized by ultralong lifetime, high-efficiency phosphorescence, and various luminescent colors. This strategy revolves around incorporating specific functional groups into the piezoelectric host skeleton to bolster molecular dipole moments, suppress non-radiative transitions, and encourage the SOC, and simultaneously constructing the corresponding guest molecules with similar structures for isostructural doping. Comprehensive studies involving photophysical investigations, in-depth single-crystal analyses, and theoretical calculations were carried out to unravel the underlying mechanism of organic *p*ML. The formation of host–guest CT exciplexes facilitates the recombination of accumulated charges and provides a suitable energy-level platform ($^1CT_E$ and $^3CT_E$) to facilitate more efficient transitions of the excited excitons from the host to the guest species and helps promote the formation of *p*ML. Significantly, the isostructural doping strategy has demonstrated its universality, as we achieved desired *p*ML outcomes across various host and guest molecules. In addition, we took initial steps toward practical applications by fabricating stress-sensing devices through a straightforward melt-casting technique. This work may facilitate the development of eco-friendly, highly efficient organic *p*ML materials with promising applications in high-resolution optical storage, pressure-sensitive lighting and display, as well as stress visualization and monitoring.

## Methods

### Materials design and syntheses

A series of *p*ML materials have been designed using structure-modified carbazole, benzoindole, and phenothiazine derivatives as isostructural hosts and guests. In these doping systems, piezoelectric molecules were utilized as the host skeleton. To augment the proportion of phosphorescence components while preserving their inherent

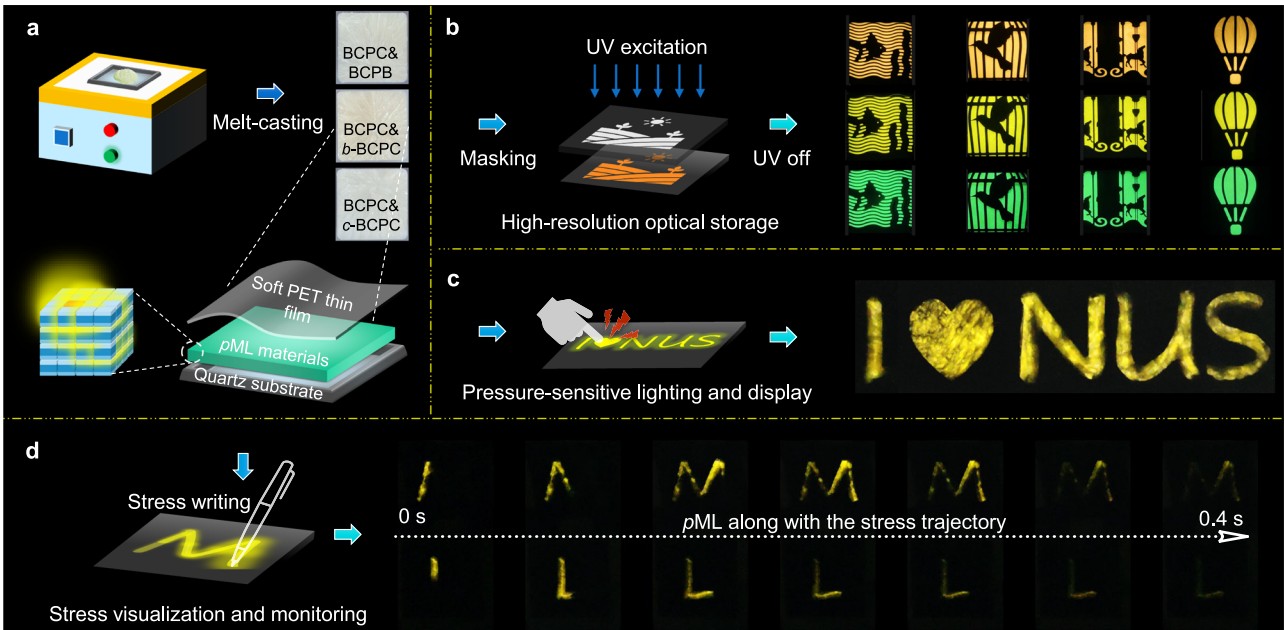

**Fig. 5 | Demonstration of applications of *p*ML materials with prolonged sensing properties. a** Prototype display and sensing devices using melt-casting method. **b**–**d** Afterglow and *p*ML applications for high-resolution optical storage (**b**), pressure-sensitive lighting and display (**c**), as well as stress visualization and monitoring (**d**).

piezoelectric effects, specific functional groups such as bromine atom and cyano group were introduced. These modifications aimed to increase the molecular dipole moments by further polarizing the piezoelectric host skeleton and suppress non-radiative transitions by enhancing intermolecular interactions. Based on this design principle, the corresponding guest molecules were constructed using a similar methodology. For syntheses, unless otherwise noted, all regents and solvents were purchased from Sigma, TCI, and BLD, and were used as received. Carbazole and all the target compounds were synthesized in accordance with the literatures[12,22,45,46]. Synthetic routes and methods for all target compounds are fully described in the Supplementary Methods.

## General methods

Proton and carbon nuclear magnetic resonance ($^1$H NMR and $^{13}$C NMR) spectra for the compounds were attained from a Bruker ARX 400 NMR spectrometer with Chloroform-$d$ (CDCl$_3$) and dimethyl sulfoxide-$d_6$ (DMSO-$d_6$) as solvent. High-resolution mass spectra (HRMS) with EI ionization were recorded on an Agilent 7200 GC-QTOF. Single-crystal X-ray analyses were collected using a Bruker D8 Venture Single-Crystal X-ray Diffractometer. PC, BPC, BCPC, and BCPSO were employed in ethyl acetate/hexane and dichloromethane/hexane mixed solvent systems to grow their single crystals utilizing the liquid-phase diffusion method. Cyclic voltammetry data were attained from an AUTOLAB PGSTAT302N instrument. The measurements were conducted in desiccated and oxygen-free dichloromethane, employing 0.1 M tetrabutylammonium hexafluorophosphate (TBAPF6) as the supporting electrolyte. A platinum wire served as the counter electrode, while a glassy carbon electrode was employed as the working electrode, and an Ag/Ag$^+$ electrode functioned as the reference electrode. Redox potentials were calibrated relative to ferrocene/ferrocenium (Fc/Fc$^+$). UV−vis absorption spectrum of each material was obtained on a UV−vis spectrometer (Shimadzu UV-2600) with their DCM solutions (50 μM). Photoluminescence (PL) spectra and delayed emission spectra were measured on an Ocean Optic QE 65 Pro spectrometer with a reflection probe R600-125F. Phosphorescence lifetimes were recorded with an Edinburgh Instruments Spectrofluorometer (FLS 1000). The phosphorescent quantum yields of the doping systems were measured by FLS 1000 Spectrofluorometer with an integrating sphere. Confocal

microscope image and 3D confocal fluorescence image were recorded using a confocal laser scanning microscopy (CLSM) (Leica TSC SP8, Germany).

## Theoretical calculations

The density functional theory (DFT) and time-dependent density functional theory (TD-DFT) calculations at B3LYP/6-311 g(d) level were performed based on single-crystal structure in Gaussian 16 program, followed by the analyzing with Multiwfn software[47,48]. The spin-orbital coupling (SOC) matrix elements between T$_1$ and S$_0$ were calculated using the ORCA program with the B3LYP functional and the DKH-def2-TZVP basic set. Crystal voids, Hirshfeld surface, and Fingerprint plot analyses base the host single crystals were conducted with the Multiwfn software. The atomic coordinates of the computational models are provided in Supplementary Data 1.

## Photocurrent measurements

Photocurrent measurements were conducted using an AUTOLAB PGSTAT302N. All samples (BCPC, BCPB, BCPC&BCPB_1%, and BCPC&BCPB_50%) were coated onto carbon papers. These materials were dissolved in a mixed solvent of tetrahydrofuran and water (v/v = 1:1) to prepare a 3.0 mM solution. Subsequently, 0.1 mL of the solution was uniformly spread onto a 1.0 cm × 1.0 cm region of the carbon paper and left to air dry. 50 mL of Na$_2$SO$_4$ aqueous solution (0.2 M) served as the electrolyte. Carbon paper, glassy carbon, and Ag/AgCl were employed as the working electrode, counter electrode, and reference electrode, respectively.

## *p*ML measurements

The *p*ML spectra were acquired using a tailor-made optical fiber probe attached to the Ocean Optic QE 65 Pro spectrometer. A quartz tube was affixed to the end of the optical fiber probe to facilitate light signal collection. Subsequently, with the spectrometer set to high-speed continuous scanning mode (with an integration time of 10 ms and a data acquisition time of 5 s), the probe promptly collided the crystalline sample and remained stationary until data collection concluded. The time zero point for the *p*ML measurements was established at the instant mechanical stimulation ceased in this work, corresponding to when the ML spectrum exhibited its maximum intensity. When

recording the *p*ML spectra at 77 K, parafilm was used to seal the bottle's mouth, preventing moisture from entering and interfering with the measurement. Subsequently, liquid nitrogen was introduced into the dewar flask to lower the temperature of the samples. The quartz bottle remained in the liquid nitrogen during the low-temperature *p*ML measurement.

### Display and stress-sensing device fabrication
Doping materials were placed within a quartz groove measuring 25 mm × 25 mm × 0.5 mm. This quartz groove was positioned on a heating stage and subjected to heating at 180 °C in the air atmosphere. After the sample had completely melted, it was removed from the heating stage and subsequently cooled to room temperature. During the cooling process, these doped materials underwent crystallization, resulting in the formation of a sleek polycrystalline panel. Lastly, a piece of polyethylene terephthalate (PET) thin film was affixed to the surface of the quartz mold using adhesive, creating a sandwich-type structure suitable for various applications in display and sensing.

### Data availability
The data that support the findings of this study are available from Z.X. and B.L. upon request. The small molecule crystallographic data for the structures reported here have been deposited at the Cambridge Crystallographic Data Centre (CCDC) under deposition numbers CCDC 2299082–2299085. These data can be obtained free of charge from the Cambridge Crystallographic Data Centre at https://www.ccdc.cam.ac.uk/structures/Search?Ccdcid=2299082-2299085&DatabaseToSearch=Published. Source data are provided with this paper.

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

## Acknowledgements

This study was supported by the Singapore National Research Foundation Investigatorship (A-8002259-00-00, B.L.), the Singapore Ministry of Education: Research Center of Excellence (A-0001423-06-00, B.L.), and the National University of Singapore (E-467-00-0032-01, B.L.). Thanks to Yi Shan for assisting in capturing the confocal fluorescence images.

## Author contributions

Z.X. and B.L. designed the project. Z.X. performed all the experiments. Z.X., Y.X. and X.Z. carried out the electrochemical and photocurrent density measurements. Z.X. and J.C. solved the crystal structures. Z.X. and Z.L. conducted the time-resolved emission spectra measurements. Z.X. and B.L. discussed the results and drafted the manuscript. B.L. supervised the project. All authors contributed to the proofreading of the manuscript.

## Competing interests

The authors declare no competing interests.
