## [Peer Review File · Nature Communications]

Isostructural Doping for Organic Persistent MechanoluminescenceREVIEWER COMMENTS

Reviewer #1 (Remarks to the Author):

In this manuscript, the authors presented isostructural doping strategy successfully constructed organic persisting mechanoluminescence (pML) materials with diverse afterglow lifetime and pML color. They elucidated the pML mechanism through multiple aspects, including the photophysical properties, single-crystal structure, theoretical simulations, and etc. They further fabricated a simple stress sensing devices through melt-casting method and conducted preliminary investigations into the application of pML.

Because this work offers a new strategy for the achievement of organic pML materials, the results are of interest and the innovation was remarkable. I suggest accept this manuscript for publication in Nature Communication after minor revision. The questions and suggestions are listed as follows:

1. In Figure S1, the delayed spectra of all hosts redshifted from ca. 450 nm to ca. 530 nm. The reason for such a large redshift was not mentioned in the manuscript or SI. Please give the reason.
2. In figure 2d, the layout of the legend easily lead readers to believe the upper figure has no ML and ML delayed 10 ms, and medium figure has no PL and PL delayed 10 ms. Suggest reorganize.
3. Since authors measured the pML spectra of the system under 77 K, please explain why the pML at 77K is weaker than that at room temperature.
4. Authors explained the pML mechanism by introducing host-guest exciplex. However, the host excimer may also lead to pML. Suggest furtherly explain.
5. To demonstrate the pML mechanism, authors introduced new hosts and guests. But the highlight of this manuscript is the isostructural doping strategy. Author should demonstrate the universality of this strategy by exchanging host and guest, that is, BCPC is as guest and BCPB is as host.
6. Because the pML was constructed based on host-guest doping strategy, authors should cite some of the classic references in this field.
7. The experimental results indicated that the introduction of CN groups played a crucial role in imparting pML performance, which was not emphasized in this manuscript. Suggest authors supplement.

Reviewer #2 (Remarks to the Author):

I have thoroughly reviewed the manuscript and found that the article is well-written, presenting a novel and significant contribution to the field of mechanoluminescence. The proposed isostructural doping strategy is supported by comprehensive experimental and theoretical analyses. The research has the potential to impact various applications, and the article effectively communicates its findings to the scientific community. However, I would like to highlight some points that, if addressed, could enhance the clarity and completeness of the manuscript:

1. Related reference should be cited after the following description” Such an excited-state process has some similarities to that in organic ML, which generally involves spatial charge separation induced by piezoelectricity.”
2. Clarify the rationale behind choosing a 1% molar percentage for the doping system. Discuss potential effects on the results if the doping ratio were to increase or decrease.
3. Justify the choice of a 1:1 ratio for photocurrent density measurements and the DFT model. Consider using the 1% doping ratio to better reflect the real situation of this work.
4. The insets of figure 2c are confusing, the color of “UV ON” for PC&PB BPC&BPB and BCPC&BCPB should be blue, blue and orange, respectively. However, they all are white or blank. Please give reasonable explanation. It also exists in Fig S20.
5. Elaborate on why molecular dipole moments are considered a key factor in the formation of organic mechanoluminescence. Provide details on how the non-centrosymmetric spatial arrangement of crystals affects ML formation and explain how dipole moments were obtained.
6. As mentioned in P10, line 203, “the introduction of bromine atom and cyano group enables the isostructural guest molecules to integrate effectively into their host matrices while significantly enhancing intermolecular interactions.” Why the PC has the highest PLQY among PC, BPC and BCPB as the BCPB has the strongest interaction and lowest non-radiative transition.
7. The detail of ML measurements and the operation of pML at 77 K should be provided. How did the author give the stress to the sample at 77K and collect the spectra simultaneously.
8. Clarify the statement regarding the substituent group -CN increasing the proportion of the phosphorescent component. Discuss whether this effect is due to its electron-absorbing nature. Additionally, explore potential differences in afterglow effects with an electron-donating substituent group.
9. For the fractured surface mentioned in P12, line 233, is there any methods to characterize it?
10. Request more evidence for the charge transfer and separation process, such as transient absorption data and information on carrier mobility.

Reviewer #3 (Remarks to the Author):

In the paper entitled “Isostructural doping for organic persistent mechanoluminescence”, the authors report on the preparation as well as photophysical characterizations, structural analysis, theoretical simulations, and mechanoluminescence (ML) properties of a series of room temperature phosphorescence host-guest systems consisting of a carbazole-based host matrix (PC, BPC, and BCPC) and an isostructural benzoindole-based guest dopant (PB, BPB, and BCPB). Finally, a mechanism for the persistent ML is proposed. The approach best on host-guest systems has already been demonstrated in the literature by at least the teams of B. Z. Tang (ref. 21) and Y. Dong (Adv. Funct. Mater. 2021, 2108072). The paper in terms of methodology, progress in the field, and results is not an impressive step forward.

These, together with a number of comments and concerns, mainly about the luminescence mechanism, do not lead me to recommend the publication of this manuscript in Nature Communications.

First, the authors characterized the 6 compounds by absorption spectra and electrochemistry to determine the HOMO and LUMO energy levels. However, the method for evaluating these values from the experimental data is not described. The authors must provide it. Moreover, the stability and reversibility of oxidized or reduced species of the studied molecules is not discussed from the voltammograms. More important, it is written (l.136) 'It was indicated that the structural similarity between host and guest molecules ensure an optimal alignment of their HOMO and LUMO energy levels, facilitating efficient charge transfer and separation within these doping systems'. What does mean an optimal alignment of the energy levels? On the other hand, it is well known in other fields of organic electronics (photovoltaics) that a difference of more than 0.3eV is necessary to avoid any reversibility of the charge separation between the donor (D) and the acceptor (A). Here, this condition is not fulfill e.g. for the LUMOs of the benzoindeole (D) and isostructural carbazole (A) derivatives. Can the authors comment? Second, the mechanism of afterglow seems analog to the one developed by Kabe et al. (ref. 32) in the case of long persistent luminescence (LPL). The mechanism lies on the fact that (i) the difference between energy levels is enough to favor charge-transfer, (ii) the acceptor radical anions can diffuse during the existence of the charge-separation state, and (iii) the donor radical cations are enough stable during the existence of the charge-separation state. These requirements are not demaonstarted in the present manuscript. Can the authors comment? Moreover, the charge recombination leads to and emission of an exciplex. It is well known that the emission band is broad and not structured. However, the delayed PL spectra are well (defined with 3 bands characteristic of the emission from a localized excited state. Can the authors comment?

Third, as for the ML mechanism, a triplet localized excited state is introduced. can the authors justify this kind of 'magic trick'?

Moreover, in the literature other mechanisms have been proposed to explain the persistent ML in host-guest systems like energy transfer (J. Yang et al. Adv. Funct. Mater. 2021, 2108072) or cluster emission (ref. 21). The authors should justify why they do not choose the later mechanism.

As for the single-crystal analysis and the relationship between crystal host matrix structure and afterglow generation, the authors must take into account the work of Demangeat et al. (Adv. Opt. Mater., 2023, 11, 2300289) on carbazole-based host matrices doped with isostructural benzoindeole-based guest derivatives. A specific 1D columnar herringbone structure favor the phosphorescence of the host-guest systems was evidenced. The authors should comment their results in light of this feature.

Besides, in the Introduction section, in Figure 1 is reported a Jablonski diagram for the excited state process of organic ML materials. However, if this scheme is known in the field of ML the authors must give the reference. If not, it must be suppress.

Regarding the photophysical properties of host, guest and doped systems (Figure 2) it seems that the caption does not match for Figure 2.d because are not reported both PL and ML for the 3 investigated systems.

The sentence (l.110) 'In comparaisn with PC&PB)' is confusing since it seems that the proportion of the phosphorescence components is directly related to the phosphorescence quantum yield. The authors must rephrase. Moreover, the phosphorescence quantum yields have to be reported in the text. It is important to note that from the ESI data, the values are not outstanding.

Corrections and changes made in response to the reviewers' comments.

Manuscript submitted to *Nature Communications*

Manuscript ID: NCOMMS-23-54365A

Title: Isostructural Doping for Organic Persistent Mechanoluminescence

We sincerely thank the reviewers for their comments about our submitted article. Corrections and necessary changes have been made according to the reviewers' comments, and are explained as follows:

(Reviewers' comments: in black; Corrections made by the authors in response to the comments: in blue)

To Reviewer #1:

Comment 1: In Figure S1, the delayed spectra of all hosts redshifted from ca. 450 nm to ca. 530 nm. The reason for such a large redshift was not mentioned in the manuscript or SI. Please give the reason.

Reply: We thank the reviewer for the valuable suggestion to improve the manuscript. As depicted in Supplementary Fig. 1, the delayed spectra of dilute toluene solutions measured at 77 K reveal four phosphorescence bands at around 411 nm, 442 nm, 467 nm, and 504 nm. In contrast, the crystalline powders measured at 77 K exhibit phosphorescence bands at around 461 nm, 475 nm, 509 nm, and 541 nm. The redshifted phosphorescence of the pure powders for the four host molecules can be assigned to aggregated phosphorescence. The redshift arises from the enhanced intermolecular interactions in the aggregated state, which is supported by the single-crystal analysis of PC, BPC, and BCPC (Supplementary Fig. 14-16). The triplet excitons are stabilized by the enhanced intermolecular interactions of these host molecules and result in the formation of

stabilized low-energy triplet excited states with redshifted aggregated phosphorescence.

In response to the reviewer' s suggestion, the reason for the redshifted phosphorescence has been incorporated into the revised Supplementary Fig. 1: "All these host molecules exhibited redshifted phosphorescence from dilute solution to the solid state. This phenomenon is attributed to the stabilization of triplet excitons by enhanced intermolecular interactions in the aggregated state, resulting in the formation of stabilized low-energy triplet excited states with redshifted aggregated phosphorescence¹."

1. Zhang X, *et al.* A Class of Organic Units Featuring Matrix-Controlled Color-Tunable Ultralong Organic Room Temperature Phosphorescence. *Adv. Sci.* **10**, 2206482 (2023).

Comment 2: In figure 2d, the layout of the legend easily lead readers to believe the upper figure has no ML and ML delayed 10 ms, and medium figure has no PL and PL delayed 10 ms. Suggest reorganize.

Reply: We appreciate the useful suggestion to improve our paper. The legends in Fig. 2d pertain to all three-part figures. To prevent potential confusion, we have consolidated the legends and updated Fig. 2d as Fig. R1 below.

Fig. R1 | PL (steady state and delayed 10 ms) and ML (prompt and delayed 10 ms) spectra of PC&PB, BPC&BPB, and BCPC&BCPB. Inset: ML images of the isostructural doping systems.

Comment 3: Since authors measured the p ML spectra of the system under 77 K, please explain why the p ML at 77K is weaker than that at room temperature.

Reply: To facilitate accurate comparison, we standardized the measurement conditions (using crystal powders of BCPC&BCPB from the same batch, maintaining equal pressure, and employing the same spectrometer parameters) while only varying the temperature for a repeated p ML measurement, and the results are shown in Fig. R2 below. It is evident that the p ML of BCPC&BCPB at 77K exhibits higher intensity and longer ML duration time compared to that at room temperature. Additionally, BCPC&BCPB displays an extra emission band (at 500 ~ 550 nm) in its p ML spectra at 77K, attributed to the phosphorescence of BCPC. These enhancements can be ascribed to the significant limitation of non-radiative transitions at low temperature.

For fair comparison, the original p ML spectra of BCPC&BCPB at room temperature and 77 K were

substituted with the new ρ ML spectra obtained under standardized conditions as Supplementary Fig. 3 and 21 in the revised Supplementary information.

Fig. R2 | TRES of ρ ML for BCPC& BCPB at room temperature (left, Supplementary Fig. 3) and 77 K (right, Supplementary Fig. 21) under standardized conditions.

Comment 4: Authors explained the ρ ML mechanism by introducing host-guest exciplex. However, the host excimer may also lead to ρ ML. Suggest furtherly explain.

Reply: For the isostructural doping systems, most of the mechanical stimulation is applied to the host molecules, and it is reasonable to infer that the host excimers may also contribute to ρ ML. In response to the reviewer' s suggestion, additional experiments were conducted to acquire time-resolved emission spectra of ρ ML for these doping systems (BCPC&BCPB, BCPC&*b*-BCPC, and BCPC&*c*-BCPC), the results are shown in Fig. R3 and R4 below. The zero-time point was established at the instant mechanical stimulation ceased in this work, corresponding to when the ML spectrum exhibited its maximum intensity. As outlined in the manuscript, the host material (BCPC) exhibited brief blue fluorescence, peaking at approximately 430 nm upon mechanical stimulation (refer to Fig. 2a), representing instantaneous ML. Notably, dual emission, comprising fluorescence from BCPC and phosphorescence from the guest species (BCPB, *b*-BCPC, or *c*-BCPC), respectively, emerged upon mechanical stimulation applied to the crystalline powders (observed in the ML spectrum at -0.01 s). Subsequently, the blue fluorescence from BCPC vanished after cessation of

mechanical stimulation, while the ultralong phosphorescence bands continued to intensify (as seen in the ML spectrum at 0.00 s). These findings suggest that the host species could indeed contribute to the generation of p ML. To improve our paper, Supplementary Fig. 22 (Fig. R3) and 25 (Fig. R4) have been added to the revised Supplementary Information. Additionally, we have added the paragraph "It is noteworthy that, the instantaneous ML emitted by the host species could be detected at the moment of mechanical stimulation applied to the crystalline powders of these isostructural doping materials (as depicted in Supplementary Fig. 22 and 25). Furthermore, the ultralong phosphorescence component continued to intensify even after the cessation of mechanical stimulation, while the blue instantaneous ML of the host species vanished. These findings suggested that the host species could also contribute to the generation of p ML." to lines 191-196 in the revised manuscript.

Fig. R3 | ML spectra of BCPC&BCPB at different collection times.

Fig. R4 | ML spectra of BCPC&*b*-BCPC and BCPC&*c*-BCPC at different collection times.

Comment 5: To demonstrate the ρ ML mechanism, authors introduced new hosts and guests. But the highlight of this manuscript is the isostructural doping strategy. Author should demonstrate the universality of this strategy by exchanging host and guest, that is, BCPC is as guest and BCPB is as host.

Reply: We appreciate the insightful question posed by the reviewer. According to the reviewer' s suggestion, we conducted experiments using BCPB as the host and BCPC as the guest to validate its feasibility. However, we found that BCPB tends to adopt an amorphous state upon aggregation. As molecules in the amorphous state are disordered, the material' s dipole moment is canceled out, making it challenging to achieve ML in BCPB&BCPC. We conducted PL, ML, and their delayed spectra measurements of BCPB&BCPC at room temperature, and the results are depicted in Fig. R5 below. The findings only revealed cyan fluorescence upon 365 nm excitation, with no afterglow emission or ρ ML detected. This underscores the critical role of the specific crystalline structure in the formation of ρ ML by providing a rigid microenvironment and facilitating the accumulation of dipole moments in the materials.

As highlighted by the reviewer, our focus in this work revolves around achieving ρ ML through the isostructural doping strategy. To verify the feasibility and universality of this approach, we systematically explored three key aspects by varying the piezoelectric skeletons and substitutional groups, as extensively detailed in the manuscript (lines 213-240). Notably, all the examined isostructural doping systems exhibited desired ρ ML with decent Φ_{Phos} and diverse ultralong lifetimes. The collective experimental results validate the feasibility and universality of the isostructural doping strategy.

Fig. R5 | PL (steady state and delayed 10 ms) and ML (prompt and delayed 10 ms) spectra of BCPB&BCPC.

Comment 6: Because the p ML was constructed based on host-guest doping strategy, authors should cite some of the classic references in this field.

Reply: To get more comprehensive citations for the host-guest doping strategy and make the article more convincing, we have added proper references [42, 43] to the sentences “Leveraging the rigid host to suppress non-radiative transition and stabilize triplet excitons, significant progress has been made in host-guest doping for achieving organic afterglow materials recently^{32-35,38-43}”.

42. Lei, Y. *et al.* Wide-Range Color-Tunable Organic Phosphorescence Materials for Printable and Writable Security Inks. *Angew. Chem. Int. Ed.* **59**, 16054-16060 (2020).

43. Ren, Y. *et al.* Clusterization-Triggered Color-Tunable Room-Temperature Phosphorescence from 1,4-Dihydropyridine-Based Polymers. *J. Am. Chem. Soc.* **144**, 1361-1369 (2022).

Comment 7: The experimental results indicated that the introduction of CN groups played a crucial role in imparting p ML performance, which was not emphasized in this manuscript. Suggest authors supplement.

Reply: The incorporation of CN groups plays a crucial role in imparting p ML performance to the

isostructural doping systems, contributing to three key aspects. First, the CN group enhances intermolecular interactions within the doping systems. These robust intermolecular interactions create a more rigid microenvironment, effectively suppressing non-radiative transitions. Second, introduction of the CN group increases the proportion of heteroatoms which can afford extra n orbitals, thereby promoting the spin-orbit coupling (SOC) between singlet and triplet states. This elevation facilitates radiative transitions between T_1 and S_0 , consequently, promoting the phosphorescence. Lastly, the CN group, functioning as an electron-absorbing moiety, effectively polarizes the host and guest molecules and enhances their charge transfer characteristics. This could result in an increased dipole moment of molecules, improving the probability of ML. Additionally, the formation of charge transfer states can reduce the lowest triplet excited state (T_1) and higher excited states (S_n or T_n , where $n \geq 2$), as depicted in Supplementary Fig. 5-7 and detailed in Supplementary Table 3-5, facilitating the transition of excited excitons, and ultimately enhancing the phosphorescence efficiency. With these synergistic effects, the introduction of the CN group demonstrates a promoting effect in imparting p ML performance to the isostructural doping systems.

To improve the quality of our manuscript, the following sentences were added to the revised manuscript:

“The cyano group, serving as an electron-absorbing moiety, can effectively polarize the host and guest molecules and enhance their charge transfer (CT) characteristics.” (lines 110-112)

“Additionally, as depicted in Supplementary Fig. 5-7 and detailed in Supplementary Table 3-5, the pronounced CT characteristics of BCPC&BCPB contributed to narrowing the energy gaps between the lowest triplet excited state (T_1) and higher excited states (S_n or T_n , where $n \geq 2$). This

reduction in energy gaps facilitated the transition of excited excitons, ultimately resulting in an enhanced phosphorescence efficiency when compared to that observed in PC&PB and BPC&BPB.” (lines 129-134)

“These findings suggest that the incorporation of the bromine atom and cyano group facilitates the effective embedding of isostructural guest molecules into host matrices, concurrently augmenting intermolecular interactions.” (lines 167-170)

“Furthermore, the introduction of bromine atom and cyano group can also enhance the spin-orbital coupling (SOC) between T_1 and S_0 (Supplementary Fig. 20) due to the heavy atom effect and heteroatom effect (affording extra n orbitals for hybrid n- π^* transition) respectively, thereby promoting phosphorescence in the doping materials.” (lines 178-182)

To Reviewer #2:

Comment 1: Related reference should be cited after the following description” Such an excited-state process has some similarities to that in organic ML, which generally involves spatial charge separation induced by piezoelectricity.”

Reply: We appreciate the careful reviewing and helpful suggestion from the reviewer. To illustrate the formation mechanism of organic ML, we have added references [1, 10] to the sentence “Such an excited-state process has some similarities to that in organic ML, which generally involves spatial charge accumulation and separation on the fractured surface induced by piezoelectricity, and then recombination to emit light^{1,10.}” .

1. Xie, Y. & Li, Z. Triboluminescence: Recalling Interest and New Aspects. *Chem* **4**, 943-971 (2018).

10. Mukherjee, S. & Thilagar, P. Renaissance of Organic Triboluminescent Materials. *Angew. Chem. Int. Ed.* **58**, 7922-7932 (2019).

Comment 2: Clarify the rationale behind choosing a 1% molar percentage for the doping system.

Discuss potential effects on the results if the doping ratio were to increase or decrease.

Reply: To elucidate the rationale behind selecting a 1% molar percentage for isostructural doping, we conducted measurements of photophysical properties for BCPC&BCPB with varying doping ratios, including steady-state and delayed emission spectra, phosphorescence lifetime, phosphorescence quantum yields (Φ_{Phos}), and afterglow performance. The results are presented in Fig. R6 and summarized in Table R1 below. It is observed that the isostructural doping system at 1% ratio exhibited the highest Φ_{Phos} and the longest phosphorescence lifetime among all BCPC&BCPB systems with different doping ratios. When the doping ratio is lower than 1%, the reduction in the formation of BCPC&BCPB exciplexes participating in afterglow emission led to a decrease in Φ_{Phos} and afterglow duration time. On the other hand, when the doping ratio is higher than 1%, an increasing number of guest molecules tend to aggregate, which induces phase separation in the doping film and quenches phosphorescence, and consequently, leading to a decrease in Φ_{Phos} and phosphorescence lifetime.

To improve the paper, we have incorporated afterglow photographs and time-resolved phosphorescence spectra of BCPC&BCPB with varying doping ratios in the Supplementary Information as Supplementary Fig. 2 (Fig. R6). Corresponding photophysical data are listed in Supplementary Table 1 (Table R1). Additionally, the discussion "Among all the isostructural doping systems with different doping ratios, BCPC&BCPB_1% exhibited the best afterglow performance with the highest ratio of phosphorescence component (R_{Phos}) and phosphorescence quantum yields (Φ_{Phos}), and the longest phosphorescence lifetime (refer to Supplementary Fig. 2 and Supplementary Table 1). Hence, all the isostructural doping systems in this work adopt the

doping ratio of 1% for experiments.” has been incorporated into the revised Supplementary Information, following Supplementary Fig. 2.

Fig. R6 | Afterglow photographs and time-resolved phosphorescence spectra of BCPC&BCPB with varying doping ratios.

Table R1. Photophysical properties of BCPC&BCPB with varying doping ratios.

Samples	λ_{PL} (nm)	λ_{Phos} (nm)	τ_{Phos} (ms)	Φ_{PL} (%)	R_{Phos} (%)	Φ_{Phos} (%)
0.1%	434, 564, 611, 665	565, 615, 671	112.9	1.46	91.4	1.33
0.5%	437, 564, 611, 665	565, 615, 671	109.6	1.76	91.7	1.61
1%	445, 564, 611, 665	565, 614, 672	114.2	2.09	91.8	1.92
5%	447, 564, 612, 665	565, 615, 671	111.6	1.09	64.2	0.70
10%	497, 564, 611, 665	565, 615, 670	101.4	0.88	66.4	0.58

Comment 3: Justify the choice of a 1:1 ratio for photocurrent density measurements and the DFT model. Consider using the 1% doping ratio to better reflect the real situation of this work.

Reply: Regarding the time-dependent density-functional theory (TD-DFT) calculations of the excited states of BCPC&BCPB, though BCPB is embedded into the BCPC matrices with a 1% doping

ratio, charge transfer occurs within the BCPC&BCPB exciplexes, while the main part of the host matrices primarily provides a rigid microenvironment for the emitting species. Consequently, employing a single host-guest pair for the TD-DFT calculations can offer a clearer depiction of the charge transfer between host and guest molecules, as well as the distribution of the excited-state energy levels.

For photocurrent density measurements, the detection of photocurrent is contingent upon the occurrence of charge transfer and separation between host and guest molecules upon photoirradiation, and the charge dispersion over long distances to the material interface. In response to the reviewer's suggestion, we recorded the photocurrent density of BCPC, BCPB, BCPC&BCPB_1%, and BCPC&BCPB_50%, with the results presented in Fig. R7 below. A clear enhancement in photocurrent density was observed in BCPC&BCPB_50%, in contrast to the marginal increase seen in BCPC&BCPB_1% compared with BCPC alone. This indicated that although charge separation within the host-guest pairs can occur in the isostructural doping system, the efficiency of charge diffusion was notably limited at a low doping ratio. Consequently, the generation of intermediate charge-separated states and subsequent gradual charge recombination are scarcely realized in these doping systems.

To improve the quality of our paper, we have updated the photocurrent density measurements and incorporated into the revised Supplementary Information as Supplementary Fig. 11 (Fig. R7). Additionally, the discussion "A clear enhancement in photocurrent density was observed in BCPC&BCPB_50%, in contrast to the marginal increase seen in BCPC&BCPB_1% compared with BCPC alone. This, combined with the non-power-law emission decay of ultralong lifetimes (refer to Fig. 2b) and small LUMO energy level differences ($\Delta E < 0.3$ eV) in between the host and guest

(see Supplementary Fig. 10), suggested that while charge separation within the host-guest pairs can indeed occur, the efficiency of charge diffusion was notably limited in these isostructural doping systems at a low doping ratio. Consequently, the generation of intermediate charge-separated states and subsequent gradual charge recombination are scarcely realized in these isostructural doping systems.” has been incorporated into the revised Supplementary Information, following Supplementary Fig. 11.

Fig. R7 | Photocurrent density variation with (on) and without (off) irradiation of a white-light source for BCPC, BCPB, and BCPC&BCPB.

Comment 4: The insets of figure 2c are confusing, the color of “UV ON” for PC&PB BPC&BPB and BCPC&BCPB should be blue, blue and orange, respectively. However, they all are white or blank. Please give reasonable explanation. It also exists in Fig S20.

Reply: The white color observed in all doping systems under “UV on” conditions was primarily resulted from the excessive intensity of the excitation light, causing the emission to become oversaturated. To avoid confusion and misunderstanding, we have re-recorded the afterglow images for all these isostructural doping systems using a more suitable intensity of UV excitation light. The luminescent photographs (inset in Fig. 2c and Supplementary Fig. S24) have been updated accordingly and shown below as Fig R8 and R9.

Fig. R8 | Time-resolved emission spectra (TRES) mapping of PC&PB, BPC&BPB, and BCPC&BCPB.

Inset: Luminescent photographs of “UV on” and “UV off” of the isostructural doping systems.

Fig. R9 | TRES mapping of BCPC&*b*-BCPC, BCPC&*c*-BCPC, BCPSO&*b*-BCPC, and BCPSO&*c*-BCPC.

Inset: Luminescent photographs of “UV on” and “UV off” of the isostructural doping systems.

Comment 5: Elaborate on why molecular dipole moments are considered a key factor in the formation of organic mechanoluminescence. Provide details on how the non-centrosymmetric

spatial arrangement of crystals affects ML formation and explain how dipole moments were obtained.

Reply: We thank the reviewer for the valuable comments. Generally, the generation of ML in many organic luminescent compounds could be explained by the electron bombardment mechanism. This mechanism involves electron bombardments occurring along with the creation of fracture surfaces, leading to charge separation and subsequent recombination to emit light. ML always accompanies the piezoelectric effect, where charge accumulation on the fracture surface intensifies electron bombardments. Consequently, piezoelectric crystals are more prone to exhibit ML. As is well-known, both large molecular dipole moments and non-centrosymmetric arrangements in crystals significantly contribute to achieving the piezoelectric property. The former can be attained by combining appropriate electron-donating groups with electron-withdrawing groups¹. The latter can be constructed by introducing polar units with small sizes into target molecules, leading to the disorder and asymmetry of molecular arrangements²⁻⁴. The dipole moment of the material equals the vector sum of all molecular dipole moments. In centrosymmetric crystals, one molecule can be completely coincident with another molecule by rotating 180° through the symmetry center, canceling out the dipole moments in crystals. On the contrary, non-centrosymmetric crystal structures facilitate the accumulation of dipole moments as molecules aggregate, contributing significantly to the piezoelectric property, as illustrated in Fig. R10 below. Hence, molecular dipole moments and non-centrosymmetric crystal structures are considered key factors to the formation of ML in organic piezoelectric crystals.

To enhance the clarity of our clarification, the sentence of "To augment the proportion of phosphorescence components while preserving their inherent piezoelectric effects, specific

functional groups such as bromine atom and cyano group were introduced. These modifications aimed to increase the molecular dipole moments by further polarizing the piezoelectric host skeleton and suppress nonradiative transitions by enhancing intermolecular interactions.” has been added to lines 379-384 in the **Methods** section of the revised manuscript.

1. Nishida J, *et al.* Phthalimide Compounds Containing a Trifluoromethylphenyl Group and Electron-Donating Aryl Groups: Color-Tuning and Enhancement of Triboluminescence. *J Org. Chem.* **81**, 433-441 (2016).
2. Li W, *et al.* Alkyl Chain Introduction: In Situ Solar-Renewable Colorful Organic Mechanoluminescence Materials. *Angew. Chem. Int. Ed.* **57**, 12727-12732 (2018).
3. Wang X, *et al.* Multicolor Ultralong Organic Phosphorescence through Alkyl Engineering for 4D Coding Applications. *Chem. Mater.* **31**, 5584-5591 (2019).
4. Xu B, *et al.* Achieving very bright mechanoluminescence from purely organic luminophores with aggregation-induced emission by crystal design. *Chem. Sci.* **7**, 5307-5312 (2016).

Fig. R10 | Dipole moments in centrosymmetric and non-centrosymmetric crystals.

Comment 6: As mentioned in P10, line 203, “the introduction of bromine atom and cyano group enables the isostructural guest molecules to integrate effectively into their host matrices while significantly enhancing intermolecular interactions.” Why the PC has the highest PLQY among PC, BPC and BCPC as the BCPC has the strongest interaction and lowest non-radiative transition.

Reply: The introduction of bromine atom and cyano group serves not only to enhance

intermolecular interactions but, more importantly, to facilitate the intersystem crossing (ISC) process by promoting spin-orbit coupling (SOC) between the singlet and triplet excited states, owing to the heavy atom effect and heteroatom effect. Consequently, singlet excited excitons in BPC and BCPC are more prone to transition to their corresponding triplet excited state, leading to the generation of additional non-radiative transition paths compared with PC. Therefore, although BCPC exhibits the strongest interaction among these three host molecules, PC still demonstrates the highest Φ_{PL} , surpassing that of BPC and BCPC. This observation holds true for doping systems PC&PB, BPC&BPB, and BCPC&BCPB. However, when comparing these three doping systems in terms of Φ_{Phos} , the trend is reversed. Since BCPC&BCPB exhibits larger SOC and stronger intermolecular interactions, it demonstrates the highest Φ_{Phos} (1.92%) among these three isostructural doping systems, compared with those of PC&PB (< 0.08%) and BPC&BPB (0.19%).

Comment 7: The detail of ML measurements and the operation of *p*ML at 77 K should be provided. How did the author give the stress to the sample at 77K and collect the spectra simultaneously.

Reply: The *p*ML spectra at 77 K were obtained using a tailor-made optical fiber probe attached to the Ocean Optic QE 65 Pro spectrometer. A round-bottom quartz tube was fixed at the end of the optical fiber probe to facilitate light signal collection and protect the fiber probe. The fiber probe was inserted into a quartz bottle containing the crystalline samples, and parafilm was applied around the bottle's mouth to prevent moisture from interfering with the measurement when liquid nitrogen was introduced into the Dewar flask to lower the temperature of the samples. Subsequently, with the spectrometer set to high-speed continuous scanning mode (with an integration time of 10 ms and a data acquisition time of 5 seconds), the probe promptly collided with the crystalline sample and remained stationary until data collection was done. The quartz

bottle was placed in the liquid nitrogen during the ρ ML measurement. The schematic diagram of the setup is presented in Fig. R11 below.

To enhance clarity, detailed information on ML measurement and the procedure for ρ ML at 77 K has been included in the **Methods** section under the subsection on ρ ML measurements (lines 429-439).

Fig. R11 | Schematic diagram of the setup for the ρ ML measurement at 77 K.

Comment 8: Clarify the statement regarding the substituent group -CN increasing the proportion of the phosphorescent component. Discuss whether this effect is due to its electron-absorbing nature. Additionally, explore potential differences in afterglow effects with an electron-donating substituent group.

Reply: we thank the reviewer for the insightful comment. The incorporation of the CN group played a crucial role in imparting ρ ML performance to the isostructural doping systems, contributing to three key aspects. First, the CN group enhanced intermolecular interactions within the doping system. These robust intermolecular interactions create a more rigid microenvironment, effectively suppressing non-radiative transitions. Second, the introduction of the CN group increased the proportion of heteroatoms which can afford extra n orbitals, thereby promoting the spin-orbit coupling (SOC) between singlet and triplet states. This elevation facilitated radiative transitions between T_1 and S_0 , consequently promoting the phosphorescence. Lastly, the CN group,

functioning as an electron-absorbing moiety, effectively polarized molecules and enhanced their charge transfer characteristics. This resulted in an increased dipole moment of molecules, improving the probability of ML. Additionally, the formation of charge transfer states can reduce the lowest triplet excited state (T_1) and higher excited states (S_n or T_n , where $n \geq 2$), as depicted in Supplementary Fig. 5-7 and detailed in Supplementary Table 3-5, facilitating the transition of excited excitons, and ultimately enhancing the phosphorescence efficiency. With these synergistic effects, the introduction of the CN group demonstrates a promoting effect in imparting p ML performance of the isostructural doping systems.

In response to the reviewer's suggestion, we designed and synthesized a new pair of host and guest molecules, named BMOPC and BMOPB, wherein the CN group was replaced with the methoxy group. The introduction of the methoxy group also provides a heteroatom effect and increases intermolecular interactions. It allows us to attribute the differences in afterglow properties more prominently to the electron-donating nature of the methoxy group. We constructed the isostructural doping system BMOPC&BMOPB and investigated its afterglow performance. The molecular structures and measurement results are depicted in Fig. R12 below. It reveals that BMOPC&BMOPB exhibit a shorter lifetime and a lower ratio of phosphorescence components (R_{Phos}) compared with BCPC&BCPB.

We have included the following sentences to the revised manuscript:

"The cyano group, serving as an electron-absorbing moiety, can effectively polarize the host and guest molecules and enhance their charge transfer (CT) characteristics." (lines 110-112)

"Additionally, as depicted in Supplementary Fig. 5-7 and detailed in Supplementary Table 3-5, the pronounced CT characteristics of BCPC&BCPB contributed to narrowing the energy gaps

between the lowest triplet excited state (T_1) and higher excited states (S_n or T_n , where $n \geq 2$). This reduction in energy gaps facilitated the transition of excited excitons, ultimately resulting in an enhanced phosphorescence efficiency when compared to that observed in PC&PB and BPC&BPB.” (lines 129-134)

“These findings suggest that the incorporation of the bromine atom and cyano group facilitates the effective embedding of isostructural guest molecules into host matrices, concurrently augmenting intermolecular interactions.” (lines 167-170)

“Furthermore, the introduction of bromine atom and cyano group can also enhance the spin-orbital coupling (SOC) between T_1 and S_0 (Supplementary Fig. 20) due to the heavy atom effect and heteroatom effect (affording extra n orbitals for hybrid $n-\pi^*$ transition) respectively, thereby promoting phosphorescence in the doping materials.” (lines 178-182)

Fig. R12 | Molecular structures of the host and guest molecules and the corresponding afterglow properties of their doping systems.

Comment 9: For the fractured surface mentioned in P12, line 233, is there any methods to characterize it?

Reply: We would like to thank the reviewer for the careful review and valuable questions. The

mechanism of organic ML is commonly understood as the electron bombardment process occurring simultaneously with the creation of fracture surfaces, leading to charge separation and subsequent recombination to emit light. In response to the reviewer's inquiry, we utilized a large-size single crystal (2 mm × 4 mm × 0.5 mm) of BCPC&*b*-BCPC to record its ML position and characterize the corresponding fracture surfaces. As illustrated in Fig. R13a, when a sharp knife scratches the crystal surface, intense *p*ML is observed solely on the scratched marks. The fracture surface is clearly visible under a confocal fluorescence microscope (see Fig. R13b). Furthermore, we conducted 3D confocal fluorescence imaging measurements to assess the distribution of mechanical stimulation applied to the crystal. As depicted in Fig. R13c, the entire crystal emitted uniform yellow fluorescence when excited by a 405 nm laser, with a more pronounced luminescence signal observed at the fracture surfaces. This observation suggests that mechanical stimulation predominantly affects the fracture surface, resulting in the in-situ generation of *p*ML. To enhance the clarity of our paper, Fig. R13 has been included in the revised Supplementary Information as Supplementary Fig. 26 with the following clarification: "Intense *p*ML is observed exclusively on the scratched marks when a sharp knife scratches the crystal surface. 3D confocal fluorescence imaging measurements were recorded to assess the distribution of mechanical stimulation applied to the crystal. The entire crystal emitted uniform yellow fluorescence when excited by a 405 nm laser, with a more pronounced luminescence signal observed at the fracture surfaces (see Supplementary Fig. 26c). It reveals that, apart from the fractured surface, the other parts of the single crystal are uniform. Combined with the *p*ML photograph and the confocal fluorescence images of the fractured surface of the BCPC&*b*-BCPC single crystal, it can be concluded that mechanical stimulation only acts on the fracture surface, generating *p*ML in situ." .

Fig. R13 | *p*ML photograph (a), confocal microscope image (b), and 3D confocal fluorescence image (c) of the fracture surface for BCPC&*b*-BCPC single crystal. Please take note that the uniform yellow fluorescent background was observed due to the optical excitation of the molecules under confocal microscope.

Comment 10: Request more evidence for the charge transfer and separation process, such as transient absorption data and information on carrier mobility.

Reply: As addressed in the response to Comment 3 from reviewer 2, we have re-examined the proposed mechanism of *p*ML. As is well known, the emission decay of organic long persistent luminescence follows a power-law decay, which indicates the generation of intermediate charge-separated states and successive gradual bulk charge recombination^{1,2}. However, the ultralong lifetimes of these isostructural doping systems exhibited exponential decay corresponding to phosphorescence, rather than the power-law emission decay observed in typical long persistent luminescence, as depicted in Fig. 2b and Supplementary Fig. 23. Moreover, results from photocurrent density measurements (as shown in Supplementary Fig. 11) indicated that although charge separation within the host-guest pairs can occur in the isostructural doping system, the efficiency of charge diffusion was notably limited at a low doping ratio. Additionally, the small differences ($\Delta E < 0.3$ eV) in lowest unoccupied molecular orbital (LUMO) energy levels between

the host and guest of these doping systems could potentially lead to the reversion of charge separation (refer to Supplementary Fig. 10). This also indirectly suggested that charge diffusion is limited in these isostructural doping systems. Consequently, the generation of intermediate charge-separated states and subsequent gradual charge recombination are scarcely realized in these doping systems.

In addition, as mentioned in response to the previous query (Comment 4 from reviewer 1), the host species could also contribute to the generation of *p*ML. Therefore, we have corrected the originally proposed *p*ML mechanism by eliminating the steps involving charge diffusion and the formation of intermediate charge-separated states, while incorporating the contribution of the host species. The revised Jablonski diagram to illustrate the updated photophysical processes of *p*ML is shown in Fig. R14 below and Fig. 3e in the revised manuscript. The discussion is added to lines 200-206 of the revised manuscript.

Regarding charge transfer, both transient absorption and emission data can serve as tools for verifying this process. However, acquiring transient absorption spectra requires transparent samples for measurement, which is challenging to achieve with the crystalline powders of our isostructural doping systems. Therefore, we conducted transient emission measurements at the nanosecond timescale to assess the charge transfer properties during the excited-state processes of ultralong phosphorescence in the isostructural doping systems. Compared with locally excited states, charge transfer excited states often exhibit a redshift in transient emission due to excited-state conformational adjustments. As shown in Supplementary Fig. 4, the introduction of the cyano group results in obvious charge transfer properties for BCPC and BCPB, with their transient emissions undergoing a redshift from 456 nm and 497 nm to 483 nm and 542 nm, respectively. In

addition, the transient emission wavelength and redshift of BCPC&BCPB (shifting from 475 nm to 493 nm) differed from those of BCPC and BCPB individually. This difference, together with the unstructured and broadened emission band, indicates the formation of a charge transfer exciplex in BCPC&BCPB.

We have included the discussion in the revised manuscript (lines 112-120) as follows: "Time-resolved emission spectra (TRES) of BCPC, BCPB, and BCPC&BCPB at the nanosecond timescale (see Supplementary Fig. 4) were recorded to investigate the CT characteristics of the isostructural doping system. The introduction of the cyano group led to evident CT properties in BCPC and BCPB, manifesting as a redshift in their transient emissions from 456 nm and 497 nm to 483 nm and 542 nm, respectively. Notably, both the transient emission wavelength and redshift of BCPC&BCPB (shifting from 475 nm to 493 nm) are different from those of BCPC and BCPB, individually. This difference, coupled with the unstructured and broadened emission bands, indicated the formation of a CT exciplex in BCPC&BCPB."

1. Kabe R, Adachi C. Organic long persistent luminescence. *Nature* **550**, 384-387 (2017).
2. Jinnai K, Kabe R, Lin Z, Adachi C. Organic long-persistent luminescence stimulated by visible light in p-type systems based on organic photoredox catalyst dopants. *Nat. Mater.* **21**, 338-344 (2022).

Fig. R14 | Jablonski diagram showing the proposed mechanism of organic p ML in isostructural doping systems with BCPC&BCPB as an example. Abbreviations: charge recombination (CR), intersystem crossing (ISC), reversed-intersystem crossing (RISC), energy transfer (ET) or charge transfer (CT) between different singlet (or triplet) excited states, and internal conversion (IC).

To Reviewer #3:

Comment 1: In the paper entitled "Isostructural doping for organic persistent mechanoluminescence", the authors report on the preparation as well as photophysical characterizations, structural analysis, theoretical simulations, and mechanoluminescence (ML) properties of a series of room temperature phosphorescence host-guest systems consisting of a carbazole-based host matrix (PC, BPC, and BCPC) and an isostructural benzoindole-based guest dopant (PB, BPB, and BCPB). Finally, a mechanism for the persistent ML is proposed. The approach best on host-guest systems has already been demonstrated in the literature by at least the teams of B. Z. Tang (ref. 21) and Y. Dong (Adv. Funct. Mater. 2021, 2108072). The paper in terms of methodology, progress in the field, and results is not an impressive step forward. These, together with a number of comments and concerns, mainly about the luminescence mechanism, do not

lead me to recommend the publication of this manuscript in Nature Communications.

Reply: We read very carefully the papers published by B. Z. Tang (Nat. Commun. 2019, 10, 5161.) and Y. Dong (Adv. Funct. Mater. 2021, 31, 2108072.). It is important to note that the host-guest doping systems reported by Y. Dong (Adv. Funct. Mater. 2021, 31, 2108072.) only exhibited instantaneous ML but not p ML. While B. Z. Tang (Nat. Commun. 2019, 10, 5161.)' s system demonstrated p ML through host-guest doping, the work was focused on electron-deficient molecules with energetically close singlet excited states as both host (ML-active) and guest components. This approach avoided any photo-induced electron transfer process but focused on the construction of cluster exciton transient states to accelerate the intersystem crossing process and achieve ultralong phosphorescence.

Our approach differs from Prof Tang' s work in terms of materials design, operation mechanism, and p ML material performance, which represents an important step forward.

We focus on piezoelectric hosts and the corresponding guests sharing similar structures for isostructural doping. Regarding the mechanism, the formation of host-guest exciplexes facilitates the recombination of accumulated charges and provides CT state (1CT_E and 3CT_E) to promote more efficient transitions of the excited excitons from host to guest, resulting in high-efficiency p ML. Furthermore, the isostructural doping approach can easily enable us to achieve a wide range of lifetime-tunable and color-tunable p ML. We have successfully synthesized a series of p ML materials with diverse phosphorescence lifetimes (ranging from 18.8 ms to 384.1 ms) and multiple p ML colors (green, yellow, and orange) at room temperature. The innovative design strategy, new mechanisms, and tunable p ML performance collectively contributed to the novelty and impact of our work.

Comment 2: The authors characterized the 6 compounds by absorption spectra and electrochemistry to determine the HOMO and LUMO energy levels. However, the method for evaluating these values from the experimental data is not described. The authors must provide it. Moreover, the stability and reversibility of oxidized or reduced species of the studied molecules is not discussed from the voltamograms. More important, it is written (l.136) ‘It was indicated that the structural similarity between host and guest molecules ensure an optimal alignment of their HOMO and LUMO energy levels, facilitating efficient charge transfer and separation within these doping systems’ . What does mean an optimal alignment of the energy levels? On the other hand, it is well known in other fields of organic electronics (photovoltaics) that a difference of more than 0.3eV is necessary to avoid any reversibility of the charge separation between the donor (D) and the acceptor (A). Here, this condition is not fulfill e.g. for the LUMOs of the benzoindole (D) and isostructural carbazole (A) derivatives. Can the authors comment?

Reply: We thank the reviewer for the valuable suggestion. As illustrated in Fig. 10, the LUMO energy levels of PB (-2.2 eV), BPB (-2.2 eV), and BCPB (-2.5 eV) were found to be higher than those of PC (-2.3 eV), BPC (-2.3 eV), and BCPC (-2.7 eV), respectively. We do agree with the reviewer that the small differences in LUMO energy levels within these doping systems could potentially hinder charge diffusion and lead to the reversion of intermediate charge-separated states. Therefore, during the revision, we conducted more experiments to valid our proposed mechanism. The non-power-law emission decay of ultralong lifetimes and the marginal increasement of photocurrent density of 1% doping ratio suggest that intermediate charge-separated states and subsequent gradual charge recombination are rarely observed in these isostructural doping systems. This agrees with the HOMO and LUMO energy level data for these host-guest pairs. Therefore, we have

revised the proposed pML mechanism accordingly by eliminating the steps involving charge diffusion and the generation of intermediate charge-separated states.

In response to the reviewer's suggestion, we have incorporated a method for evaluating the E_{HOMO} and E_{LUMO} values from the experimental data and a clarification for HOMO and LUMO of these isostructural host-guest pairs into the revised Supplementary Information. This addition can be found following Supplementary Fig. 10: "The values of the highest occupied molecular orbital (HOMO) and lowest unoccupied molecular orbital (LUMO) energy levels for these host and guest molecules can be calculated from the absorption and cyclic voltammetry data, as per the formulas (1)-(3). Here, E^{ox} represents the oxidation onset potential, E_g is the energy gap of the HOMO and LUMO within a molecule, and λ_{abs} is the initial absorption wavelength from the long-wavelength region. The small differences ($\Delta E < 0.3$ eV) in lowest unoccupied molecular orbital (LUMO) energy levels between the host and guest of these doping systems could potentially lead to the reversion of charge separation^{2,3}. This indirectly suggested that charge diffusion is limited in these isostructural doping systems." We express our appreciation to the reviewers for their thorough examination and insightful queries regarding the luminescence mechanism, which have significantly contributed to the improvement of our paper.

2. Tamai Y, Shirouchi R, Saito T, Kohzuki K, Natsuda S-i. Role of the energy offset in the charge photogeneration and voltage loss of nonfullerene acceptor-based organic solar cells. *J. Mater. Chem. A* **11**, 17581-17593 (2023).

3. Jasiūnas R, Zhang H, Gelžinis A, Chmeliov J, Franckevičius M, Gao F, Gulbinas V. Interplay between charge separation and hole back transfer determines the efficiency of non-fullerene organic solar cells with low energy level offset. *Org. Electron.* **108**, 106601 (2022).

Comment 3: The mechanism of afterglow seems analog to the one developed by Kabe et al. (ref. 32) in the case of long persistent luminescence (LPL). The mechanism lies on the fact that (i) the difference between energy levels is enough to favor charge-transfer, (ii) the acceptor radical anions can diffuse during the existence of the charge-separation state, and (iii) the donor radical cations are enough stable during the existence of the charge-separation state. These requirements are not demonstrated in the present manuscript. Can the authors comment? Moreover, the charge recombination leads to and emission of an exciplex. It is well known that the emission band is broad and not structured. However, the delayed PL spectra are well (defined with 3 bands characteristic of the emission from a localized excited state. Can the authors comment?

Reply: As discussed in our responses to previous queries (Comment 10 from reviewer 2 and Comment 2 from reviewer 3), we have revised our initially proposed *p*ML mechanism based on analyses of the results regarding ultralong lifetimes, photocurrent density, and HOMO/LUMO energy levels for these isostructural doping systems. The updated *p*ML mechanism, which no longer includes steps involving charge diffusion and the generation of intermediate charge-separated states (evidenced by the non-power-law emission decay of ultralong lifetimes), has been integrated into the revised manuscript as Fig. 3e, along with corresponding descriptions added to lines 206-212.

Additionally, although these isostructural doping systems have different excitation pathways, it can be concluded from their PL and ML spectra that both the photoexcited afterglow and *p*ML undergo the same radiative transition pathways, emitting from the lowest triplet locally excited states of guest molecules (3LE_G). To avoid misunderstanding and enhance clarity in our paper, the paragraph elucidating the formation of afterglow emission has been rephrased and incorporated

into lines 109-134 of the revised manuscript.

Comment 4: As for the ML mechanism, a triplet localized excited state is introduced. Can the authors justify this kind of 'magic trick'? Moreover, in the literature other mechanisms have been proposed to explain the persistent ML in host-guest systems like energy transfer (J. Yang et al. Adv. Funct. Mater. 2021, 2108072) or cluster emission (ref. 21). The authors should justify why they do not choose the later mechanism.

Reply: As outlined in response to comment 3 above, both the photoexcited afterglow and *p*ML follow the same radiative transition pathways, emitting from the lowest triplet locally excited states of guest molecules (³LE_G).

Regarding the explanation of *p*ML mechanism, it's crucial to note that the host-guest doping systems reported by J. Yang et al. (Adv. Funct. Mater. 2021, 31, 2108072.) solely exhibit instantaneous ML rather than the *p*ML we focused on. In B. Z. Tang (Nat. Commun. 2019, 10, 5161.)'s work, the authors proposed the cluster emission mechanism for *p*ML, wherein a cluster exciton spanning the host and guest forms as a transient state to accelerate the intersystem crossing process and produce ultralong phosphorescence. They exclusively selected electron-deficient molecules with energetically close singlet excited states as both host (ML-active) and guest components to prevent any photo-induced electron transfer but enable cluster exciton formation. We are interested in developing a more general strategy to realize *p*ML and therefore we proposed the isostructural doping systems, which are not limited to cluster emission. Our system does not operate on cluster emission for the following reasons: firstly, the substantial energy gaps between the singlet excited states of the guest and the host (0.5 eV for PC&PB and BPC&BPB, 0.8 eV for BCPC&BCPB) make it difficult to form cluster excitons in these systems.

Furthermore, the piezoelectric hosts and corresponding isostructural guests are not electron-deficient molecules; instead, photo-induced electron transfer (intra- and intermolecular charge transfer) has been demonstrated to be pivotal in the formation of high-efficiency pML . Accordingly, we proposed the charge transfer exciplex mechanism for the isostructural doping-induced pML . This mechanism involves the generation of the host-guest CT exciplex, which can efficiently facilitate charge recombination and establish a suitable energy level platform (1CT_E and 3CT_E) between the host and guest species, facilitating more efficient transitions of the excited excitons from higher host energy levels to the lowest triplet states (3LE_G), ultimately emitting high-efficiency ultralong phosphorescence to achieve pML . The materials design and the mechanism clearly demonstrate the novelty of our work.

Comment 5: As for the single-crystal analysis and the relationship between crystal host matrix structure and afterglow generation, the authors must take into account the work of Demangeat et al. (Adv. Opt. Mater., 2023, 11, 2300289) on carbazole-based host matrices doped with isostructural benzoindole-based guest derivatives. A specific 1D columnar herringbone structure favor the phosphorescence of the host-guest systems was evidenced. The authors should comment their results in light of this feature.

Reply: In the study by Demangeat et al. (Adv. Opt. Mater., 2023, 11, 2300289), the authors designed a series of carbazole derivatives and their isomers with D- σ -A structures to construct host-guest doping systems and emphasized the importance of a specific 1D columnar herringbone structure composed of a carbazole (Cz) stacks pair for achieving organic room-temperature phosphorescence. However, while crystallization significantly influences the formation of room-temperature phosphorescence, it is not a necessary factor, let alone a specific crystalline pattern.

As exemplified by BCPC&BCPB, we can produce an amorphous doping film through rapid annealing from the molten state, resulting in an orange afterglow lasting 1 second after the cessation of UV light at room temperature, as depicted in Fig. R15 below. This phenomenon is also observed in some other isostructural doping systems in our group. Consequently, it can be concluded that crystallization is not a requisite factor to the generation of afterglow in isostructural doping systems. This represents another key advantage of our designed system.

Fig. R15 | Afterglow photographs and time-resolved phosphorescence spectra of BCPC&BCPB in amorphous state.

Comment 6: In the Introduction section, in Figure 1 is reported a Jablonski diagram for the excited state process of organic ML materials. However, if this scheme is known in the field of ML the authors must give the reference. If not, it must be suppress.

Reply: We appreciate the valuable suggestion from the reviewer to enhance the quality of our paper. It is imperative to highlight that not all organic ML processes follow the same excited-state pathways as photoluminescence (PL). In certain organic ML molecules, phosphorescence can't be detectable in their PL spectra but becomes evident in the ML process¹. The ML excitation process, unlike the PL excitation process, does not involve the absorption of a photon and is thus not necessarily a vertical electronic transition². Instead, ML is akin to an emission arising from the recombination of an exciton (Frenkel and Wannier exciton) in molecules³. However, in some cases,

the formation of ML is attributed to the excitation of N₂ under mechanical stimulation, subsequently exciting the ML molecules. This scenario falls under the category of photo-excitation processes. As a result, Fig. 1b has been excluded from the revised manuscript to prevent misinterpretation, as it does not comprehensively depict the excited state processes of all organic ML materials.

1. Xie, Y. & Li, Z. Triboluminescence: Recalling Interest and New Aspects. *Chem* **4**, 943-971 (2018).
2. Zink JI. Triboluminescence. *Acc. Chem. Res.* **11**, 289-295 (1978).
3. Yang J, *et al.* AI Egen with Fluorescence–Phosphorescence Dual Mechanoluminescence at Room Temperature. *Angew. Chem. Int. Ed.* **56**, 880-884 (2017).

Comment 7: Regarding the photophysical properties of host, guest and doped systems (Figure 2) it seems that the caption does not match for Figure 2.d because are not reported both PL and ML for the 3 investigated systems.

Reply: The legends in Fig. 2d pertain to all three-part figures. To prevent potential confusion, we have consolidated the legends and updated the figure as Fig. R16 below.

Fig. R16 (Fig. 2d) | PL (steady state and delayed 10 ms) and ML (prompt and delayed 10 ms) spectra of PC&PB, BPC&BPB, and BCPC&BCPB. Inset: ML images of the doping systems.

Comment 8: The sentence (l.110) ‘In comparison with PC&PB)’ is confusing since it seems that the proportion of the phosphorescence components is directly related to the phosphorescence quantum yield. The authors must rephrase. Moreover, the phosphorescence quantum yields have to be reported in the text. It is important to note that from the ESI data, the values are not outstanding.

Reply: In response to the helpful suggestion from the reviewer, the previous sentence has been revised and incorporated into lines 90-95 in the revised manuscript: “In comparison with PC&PB, the proportion of phosphorescence components of the other two doping materials, BPC&BPB and especially BCPC&BCPB, experienced substantial enhancements, leading to a remarkable increase of over 90-fold in the ratio of phosphorescence component (R_{Phos}), as detailed in Supplementary

Table 2. The phosphorescence quantum yields (Φ_{Phos}) of BCPC&BCPB (1.92%) also exhibited a significant enhancement compared with PC&PB (< 0.08%), as shown in Fig. 2b." .

The Φ_{Phos} values of these three isostructural doping systems are also presented in Fig. 2b. While the Φ_{Phos} values for these three isostructural doping systems are not very high, with the highest being 1.92% for BCPC&BCPB, these systems serve as prototype doping systems to demonstrate the feasibility and rationality of the design strategy. Building on this design principle of isostructural doping, we have also created several new isostructural doping systems with improved afterglow properties, featuring the longest lifetime of 384.1 ms and the highest Φ_{Phos} of 11.89%, as detailed in the manuscript and Supplementary Table 2.

REVIEWERS' COMMENTS

Reviewer #1 (Remarks to the Author):

I read the revised manuscript. I thought that authors carefully revised and modified their manuscript according to the referees' comments by point-to-point. So that, I suggest accept this manuscript for publication in Nature Communication.

Reviewer #2 (Remarks to the Author):

The Author has addressed all the concerns I have, and the quality of this paper has improved greatly. Therefore, I recommend its acceptance in this journal.

Reviewer #3 (Remarks to the Author):

The manuscript has been further improved in the rebuttal, addressing satisfactorily all issues raised in the first round of revision. As such, the manuscript now appears to much better fit the standards for publication in Nature Communications, and I can thus recommend it for publication.